

# Genetic structure and designing a preliminary core collection of *Zizania latifolia* in China based on 12 microsatellites markers

Xiangliang Lei[1,2,*], Xiaona Su[3,*], Chengchuan Zhou[4], Shaolin Jiang[1], Xiaoquan Yuan[2], Yao Zhao[5] and Shaomei Jiang[6]

[1] Key Laboratory of Crop Physiology, Ecology and Genetic Breeding, Jiangxi Agricultural University, Nanchang, Jiangxi, China
[2] Fuzhou Institute of Agricultural Science, Fuzhou, Jiangxi, China
[3] Nanchang Business College, Jiangxi Agricultural University, Jiujiang, Jiangxi, China
[4] Jiangxi Academy of Forestry, Nanchang, Jiangxi, China
[5] Key Laboratory of Poyang Lake Environment and Resource Utilization, Ministry of Education, Center for Watershed Ecology, School of Life Sciences, Nanchang University, Nanchang, Jiangxi, China
[6] School of Statistics and Data Science, Jiangxi University of Finance and Economics, Nanchang, Jiangxi, China
* These authors contributed equally to this work.

Corresponding authors
Yao Zhao, yaozhao@ncu.edu.cn
Shaomei Jiang,
13870621151@163.com

## ABSTRACT

The genetic diversity and structure of wild crop relatives are crucial for their conservation and utilization in breeding programs. This study presents a comprehensive survey and collection of *Zizania latifolia* across its natural distribution range in China. Using 12 microsatellite markers, the genetic diversity of 357 wild *Z. latifolia* accessions from 25 populations was evaluated, revealing a high genetic diversity ($H_e$ = 0.439). The genetic structure analysis indicated significant genetic differentiation among populations, with evidence of isolation by distance. CoreHunter3 and PowerMarker software were employed to design a preliminary core collection, and the final core collection comprised 92 wild accessions. The core collection was found to be representative of the original germplasm, ensuring the effective conservation of *Z. latifolia*'s genetic resources. This study would provide valuable insights for the development of conservation strategies and the utilization of *Z. latifolia*.

## INTRODUCTION

The emergence and advancement of agricultural civilization has laid the fundamental material foundation for the development and progress of human society. Despite the substantial growth in cultivated plant production over the past century, the diversity of cultivated plants has declined significantly, particularly among major food crops (*Meyer, DuVal & Jensen, 2012*; *Isbell et al., 2017*). The three primary staple crops—wheat, maize,

and rice—currently occupy approximately 80% of the global cereal acreage (*Food and Agriculture Organization (FAO), 2022*). Crop wild relatives play an important role in crop breeding and food security, as they possess far greater genetic diversity and superior agronomic and resistance traits that are often lacking in cultivated varieties (*Zhang et al., 2017*). Exploring and utilizing the genetic resources of wild crop relatives to improve existing crops, breed new high-yielding and high-resistance varieties, or even domesticate novel crops from wild plants, is considered one of the most effective approaches to address future challenges and achieve sustainable societal development (*Choudhary et al., 2017*; *Kashyap et al., 2022*).

The survey, collection, and evaluation of crop wild relatives form the foundation for their conservation and utilization. However, the conservation of crop wild relatives in large quantities, while providing a rich genetic base for plant genetic improvement and variety selection, also presents challenges in terms of physical space and maintenance costs (*Brush & Meng, 1998*; *Ford Lloyd et al., 2011*). To address this problem, *Frankel (1984)* first proposed the concept of core germplasm, which was further developed and refined by *Brown (1989)*. The identification and conservation of core germplasm have been widely adopted in cultivated plants, such as rice, sweet potato and olive (*Belaj et al., 2012*; *Su et al., 2017*; *Kumar et al., 2020*). However, relatively few efforts have been made to create core germplasm for wild plants, especially wild relatives of crops (*Miyamoto, Ono & Watanabe, 2015*; *Bai et al., 2019*; *Wang et al., 2023*), which evidently limits our ability to further explore and utilize their genetic resources.

The genus *Zizania* taxonomically belongs to the family Gramineae, subfamily Rice, and tribe Rice (*Chen & Xu, 1994*; *Terrell et al., 1997*). Globally, there are four recognized *Zizania* species: *Z. latifolia*, *Z. palustris*, *Z. aquatica*, and *Z. texana*. While *Z. latifolia* is found in East Asia, the latter three species are native to North America. *Z. palustris*, commonly known as wild rice in North America, was traditionally consumed as a food by indigenous communities and subsequently domesticated as a specialty food crop in the region. Similarly, *Z. latifolia* was once utilized as a food crop in China, but its ancestors were eventually domesticated to produce a distinct aquatic vegetable called "Jiaobai" in the form of a plant-fungus complex (*Guo et al., 2007*; *Guo et al., 2015*; *Zhao et al., 2019*).

Today, cultivated *Z. latifolia* is a highly popular aquatic vegetable in southern China, second only to lotus root in terms of cultivation area (*Guo et al., 2007*). As a tertiary gene pool of Asian cultivated rice, *Z. latifolia* represents an important genetic resource for rice improvement, as its life history and phenotypic traits are highly analogous to those of rice, while possessing numerous unique and desirable characteristics that allow for hybridization with rice (*Liu, Liu & Li, 1999*; *Guo et al., 2015*). Inspired by the successful domestication of *Z. palustris* in North America, coupled with the development of advanced genomic breeding technologies and the acknowledged high nutritional value of *Z. latifolia*, some researchers have proposed the possibility of *de novo* domestication of *Z. latifolia* as a grain crop (*Zhao et al., 2018*; *Yan et al., 2022*; *Xie et al., 2023*).

The natural distribution range of *Z. latifolia* in China spans a wide latitudinal expanse, with wild populations extending from the southwestern to the northeastern regions, where it is a common aquatic plant in freshwater wetland habitats. However, in recent decades,

the natural habitat of *Z. latifolia* has been extensively lost due to rapid population growth, intensified human economic activities, and accelerated urbanization, leading to the decline or even local extinction of some populations. Consequently, a comprehensive survey, collection, and evaluation of wild *Z. latifolia* populations, as well as the establishment of a dedicated germplasm resource nursery, are not only necessary for the conservation of its genetic resources but also form the foundation for the exploration and utilization of this species' potential. Several studies have been conducted to investigate the genetic diversity and population genetic structure of *Z. latifolia*. For instance, *Xu et al. (2008)* analyzed 21 wild *Z. latifolia* populations using a single nuclear gene (Adh1) sequence and found low levels of nucleotide diversity, with no evident geographic population structure detected. In a subsequent study, *Chen et al. (2012*, *2017)* evaluated the genetic diversity of *Z. latifolia* populations in the middle reaches of the Yangtze River and the northeastern region of China using SSR markers, while *Zhao et al. (2018)* also examined *Z. latifolia* populations in the eastern region of China using the same molecular approach and reported a clinal genetic structure from north to south. *Zou et al. (2024)* also found two genetic groups corresponding to the north and south of *Z. latifolia*'s distribution range using whole genome resequencing. More recently, *Wagutu et al. (2022)* analyzed 28 *Z. latifolia* populations from five major watersheds using 46 SSR loci and concluded that the species exhibits relatively lower overall genetic diversity and significant genetic differentiation among watersheds. Although the findings of these studies may vary depending on the sampling strategies or methodologies employed, they collectively reveal patterns of genetic differentiation within the existing *Z. latifolia* populations and provide a solid foundation for the collection and identification of genetic resources.

Building upon the findings from previous studies, we conducted a comprehensive survey and collection of *Z. latifolia* samples across its potential natural distribution range in China, and subsequently established an *ex situ* germplasm bank for this species. We then proceeded to evaluate the genetic diversity of the *Z. latifolia* germplasm resources using SSR molecular markers and attempted to construct a preliminary core germplasm subset. This multifaceted investigation can serve as a valuable reference for the conservation and utilization of *Z. latifolia*'s genetic resources. Furthermore, the study has helped to elucidate the endangered status of wild *Z. latifolia* populations, thereby informing the development of appropriate conservation strategies for this important species.

## MATERIALS AND METHODS

### Sample collection strategy

From 2019 to 2021, we conducted surveys and collected samples based on existing digital herbarium records (Chinese Virtual Herbarium, https://www.cvh.ac.cn) at 40 potential distribution areas of *Z. latifolia* (including habitats such as lakes, rivers, and swamps) across 28 provinces. Among these areas, wild *Z. latifolia* was found in only 25 survey areas, predominantly concentrated east of the Heihe-Tengchong line (Fig. 1). In each survey area, we selected a number of evenly distributed sampling sites according to the actual size of the area, collecting three live plants from each site, with a spacing of not less than 1,000 m between individual plants. These plants were then transplanted to the aquatic plant

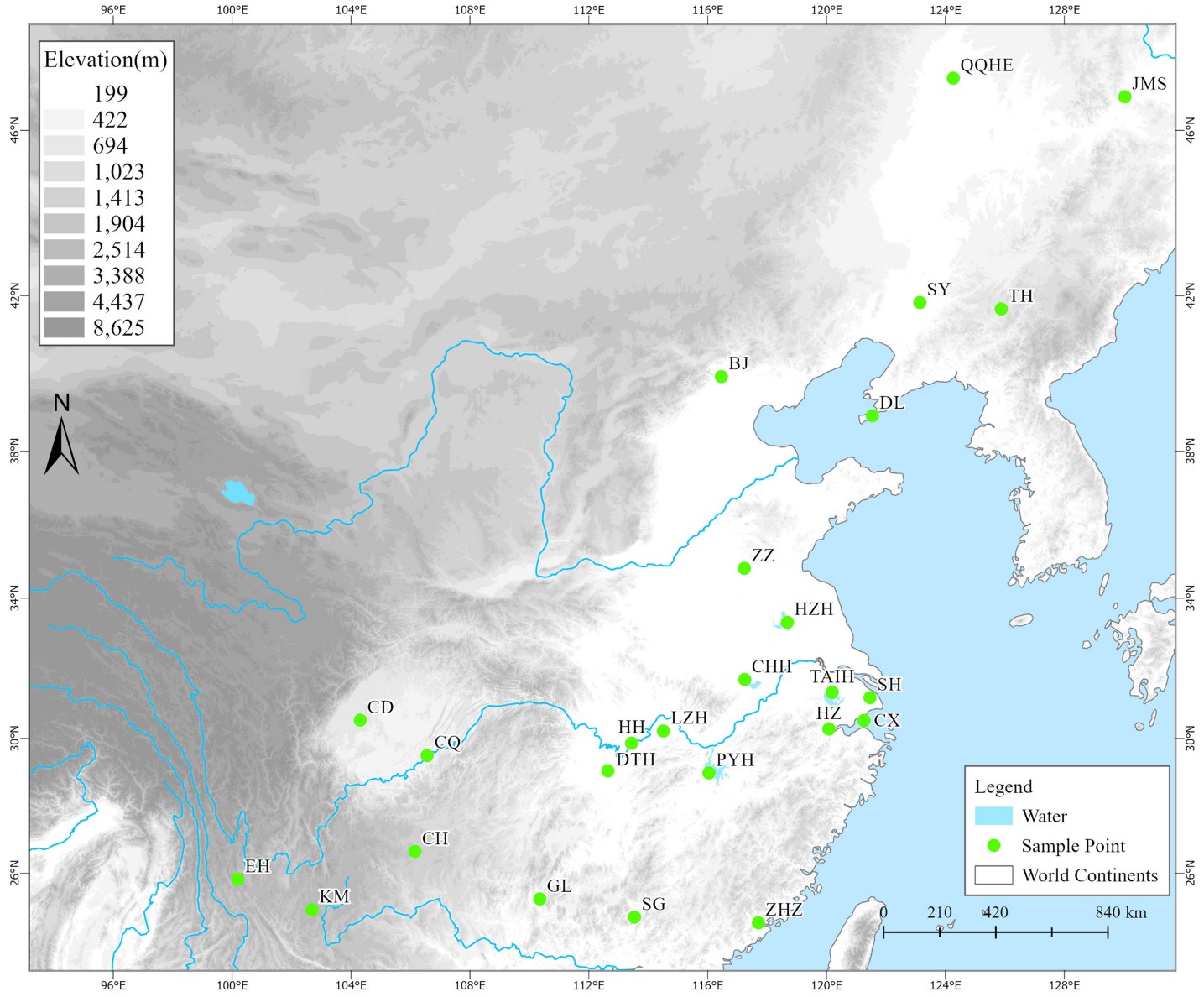

**Figure 1 The geographical locations of 25 wild *Zizania latifolia* populations in China.** A total of 25 survey localities from different areas in China. (1) **QQHE**, Qiqihar; (2) **JMS**, Jiamusi; (3) **TH**, Tonghua; (4) **SY**, Shenyang; (5) **DL**, Dalian; (6) **BJ**, Beijing; (7) **ZZ**, Zaozhuang; (8) **HZH**, Hongze; (9) **TAIH**, Wuxi; (10) **SH**, Shanghai; (11) **CX**, Cixi; (12) **HZ**, Hangzhou; (13) **CHH**, Chaohu; (14) **PYH**, Poyang; (15) **HH**, Honghu; (16) **LZH**, Wuhan; (17) **DTH**, Yueyang; (18) **CD**, Chengdu; (19) **CQ**, Chongqing; (20) **ZHZ**, Zhangzhou; (21) **SG**, Shaoguan; (22) **GL**, Guilin; (23) **CH**, Weining; (24) **KM**, Kunming; (25) **EH**, Dali. Green filled circles represent survey localities. The base map was obtained from the National Catalogue Service for Geographic Information (http://www.webmap.cn).

resource nursery at the Lushan Botanical Garden in Jiujiang, Jiangxi Province, for *ex situ* conservation. Once the collected plants were established and had survived, 3 to 5 healthy, young leaves were selected, placed in a sealed bag with an appropriate amount of color-changing silica gel, and stored in a refrigerator at −80 °C after complete drying for DNA extraction. Ultimately, we obtained a total of 357 wild *Z. latifolia* samples from 132 sampling sites across 25 populations (survey areas).

## DNA extraction and PCR assays

Total genomic DNA was extracted using a plant genomic DNA kit (Tiangen Biotech, Beijing, China). DNA quality and concentration were then examined using NanoDrop ND2000 spectrophotometer (Thermo Fisher Scientific, Waltham, MA, USA). Twelve microsatellites with clear bands and relatively high polymorphism were selected from 24 *Z. latifolia*-specific and cross-specific microsatellites after the pre-experiment, and the specifically developed markers were generally better than the cross-specific microsatellites (*Quan et al., 2009*; *Richards et al., 2004*, *2007*). PCR amplification was performed following the protocol by *Quan et al. (2009)*, and the PCR products were separated on a 6% denaturing polyacrylamide gel (Fig. S1). The fragments were then visualized by silver staining, and alleles were scored in reference to a DNA marker (100 bp ladder; Tiangen Biotech, Beijing, China). To ensure the high quality of the codominant SSR dataset, several precautions were taken, as suggested by *Bonin et al. (2004)*. Then we manually converted the raw data into the input format of the software GENALEX 6.5 (*Peakall & Smouse, 2012*).

## Genetic data analyses

The parameters of population genetic variation were estimated using mean allele number ($N_a$), mean effective allele number ($A_e$), Shannon information index ($I$), polymorphism information content ($PIC$), expected heterozygosity ($H_e$), observed heterozygosity ($H_o$), and Wright's fixation index ($FI$) with GENALEX 6.5 and PowerMarker v3.25 (*Liu & Muse, 2005*; *Peakall & Smouse, 2012*). *F*-statistics were calculated to estimate the extent of genetic differentiation among populations. Analysis of Molecular Variation (AMOVA) was conducted to examine genetic variation both among and within populations. A Principal Coordinates Analysis (PCoA) was then performed to illustrate the pattern of genetic divergence. *F*-statistics, AMOVA, and PCoA analyses were all carried out using GENALEX 6.5 (*Peakall & Smouse, 2012*). The Unweighted Pair Group Method with Arithmetic Mean (UPGMA) dendrograms of populations and individuals based on Nei's distance were respectively constructed and visualized in MEGA X (*Kumar et al., 2018*). The genetic structure was further explored using the Bayesian clustering algorithm implemented in STRUCTURE 2.3.4 (*Pritchard, Stephens & Donnelly, 2000*). The program was provided with no prior information on ancestral populations and was run 10 times for each value of *K* (*K* ranges from 1 to 25), under the admixture model with independent allele frequencies, using 1,000,000 Markov chain Monte Carlo iterations and a burn-in of 100,000 iterations. We inferred *K* using the *ad hoc* statistic $\Delta K$ (*Evanno, Regnaut & Goudet, 2005*). The resulting matrices of estimated cluster membership coefficients (Q) were permuted with CLUMPP (*Jakobsson & Rosenberg, 2007*). The final matrix for each *K* value was visualized with DISTRUCT (*Rosenberg, 2004*). Isolation by Distance (IBD) among populations was tested using the Mantel test implemented in GENALEX 6.5. The significance of IBD values was assessed using 9,999 permutations.

## Design the core collection

The core collection of *Z. latifolia* germplasm was constructed using the following methods: First, the program CoreHunter3 (*De Beukelaer, Davenport & Fack, 2018*) was employed to develop independent core collections based on genotypic data. Various cutoff values, including 10%, 15%, 20%, 25%, 30%, 40%, and 50% of the initial collection, were used to design the core collections in CoreHunter3, based on Modified Rogers distance (MR) with default parameters. Next, PowerMarker v3.25 software was utilized, and core collections were constructed using the simulated annealing algorithm (SA), specifically with maximizing allelic richness (SANA) and maximizing genetic diversity (SAGD). The cutoff values were set identically to those used in CoreHunter3. Finally, the core collections extracted by each method with the 10% cutoff value were combined to form the final core germplasm of *Z. latifolia*.

# RESULTS

## Genetic diversity

The genetic variation among 357 wild *Z. latifolia* accessions was estimated using 12 SSR markers. Generally, all 12 SSR loci were polymorphic, each generating two to 12 alleles, indicating relatively low SSR variation. The mean number of alleles ($N_a$), mean number of effective alleles ($A_e$), polymorphism information content (*PIC*), Shannon information index (*I*), observed heterozygosity ($H_o$), and expected heterozygosity ($H_e$) for each SSR locus are listed in Table S1. The values for heterozygosity varied among these loci ($H_o$ = 0–0.101, $H_e$ = 0.028–0.373). The *Z. latifolia* populations exhibited a relatively high level of mean genetic diversity, with $N_a$ = 1.823, $A_e$ = 1.443, *I* = 0.345, $H_o$ = 0.012, and $H_e$ = 0.207.

The sampling information, population size, and genetic diversity parameters for each population are listed in Table 1. The overall genetic diversity of wild *Z. latifolia* accessions used in this study was high, with $N_a$ = 5.000, $A_e$ = 1.963, $H_o$ = 0.013, and $H_e$ = 0.439. However, three populations (SH, HZ, and CD) showed no SSR variability, and all samples within these populations were genetically uniform. The unexpectedly high mean fixation index (*FI* = 0.976, Table 1) indicated significant heterozygosity deficiency, although these values varied among populations.

## Genetic structure

The *F*-statistics indicated high genetic differentiation among populations, with an overall genetic differentiation coefficient $F_{st}$ = 0.452 (Table S1). This was further confirmed by AMOVA, which revealed that 60.35% of the diversity occurred within individuals, while 37.11% was attributed to genetic differentiation among groups (Table S2).

A model-based Bayesian approach was employed to infer the genetic structure of individuals under *K* = 2, which resulted in the best fit to the model as estimated by Evanno's Δ*K* statistics (Fig. S2). According to the genetic assignment analysis of STRUCTURE, when *K* = 2, the populations of wild *Z. latifolia* could be divided into two genetic groups. Populations from high latitudes (QQHE, JMS, TH, SY, BJ, ZZ, and HZH) were clustered into Group I (North group), while the remaining populations were

**Table 1 Parameters of genetic diversity of 25 wild Z. latifolia populations based on 12 microsatellites.**

| Population code | Location | Latitude | Longitude | Population size | Sampling site number | Accession number | $N_a$ | $A_e$ | $H_o$ | $H_e$ | FI |
|---|---|---|---|---|---|---|---|---|---|---|---|
| QQHE | Qiqihar | 47.2 | 124.25 | S | 4 | 16 | 1.750 | 1.383 | 0.024 | 0.201 | 0.894 |
| JMS | Jiamusi | 46.75 | 130.08 | S | 8 | 23 | 2.667 | 2.080 | 0.037 | 0.381 | 0.903 |
| TH | Tonghua | 41.65 | 125.87 | S | 5 | 23 | 1.833 | 1.486 | 0.048 | 0.261 | 0.890 |
| SY | Shenyang | 41.81 | 123.13 | S | 6 | 21 | 2.083 | 1.661 | 0.000 | 0.337 | **1.000** |
| DL | Dalian | 38.91 | 121.46 | S | 3 | 15 | 1.750 | 1.283 | 0.000 | 0.172 | **1.000** |
| BJ | Beijing | 39.95 | 116.45 | S | 4 | 12 | 1.917 | 1.396 | 0.000 | 0.225 | **1.000** |
| ZZ | Zaozhuang | 34.81 | 117.23 | S | 3 | 9 | 1.833 | 1.661 | 0.037 | 0.289 | 0.863 |
| HZH | Hongze | 33.31 | 118.66 | L | 17 | 32 | 3.333 | 2.037 | 0.016 | 0.431 | 0.970 |
| TAIH | Wuxi | 31.31 | 120.19 | L | 4 | 12 | 2.750 | 1.936 | 0.064 | 0.464 | 0.869 |
| SH | Shanghai | 31.19 | 121.45 | S | 1 | 3 | 1.000 | 1.000 | 0.000 | 0.000 | – |
| CX | Cixi | 30.5 | 121.25 | S | 3 | 9 | 1.167 | 1.054 | 0.000 | 0.044 | **1.000** |
| HZ | Hangzhou | 30.26 | 120.06 | S | 3 | 9 | 1.000 | 1.000 | 0.000 | 0.000 | – |
| CHH | Chaohu | 31.67 | 117.23 | L | 5 | 15 | 1.333 | 1.196 | 0.000 | 0.094 | **1.000** |
| PYH | Poyang | 28.98 | 116.03 | L | 4 | 11 | 2.583 | 1.860 | 0.015 | 0.420 | 0.953 |
| HH | Honghu | 29.86 | 113.43 | L | 3 | 9 | 1.250 | 1.091 | 0.000 | 0.060 | **1.000** |
| LZH | Wuhan | 30.21 | 114.5 | L | 6 | 16 | 2.500 | 1.952 | 0.000 | 0.428 | **1.000** |
| DTH | Yueyang | 29.03 | 112.63 | L | 9 | 27 | 3.250 | 2.169 | 0.015 | 0.454 | 0.900 |
| CD | Chengdu | 30.51 | 104.3 | S | 3 | 9 | 1.000 | 1.000 | 0.000 | 0.000 | – |
| CQ | Chongqing | 29.5 | 106.55 | S | 8 | 24 | 1.583 | 1.120 | 0.000 | 0.090 | **1.000** |
| ZHZ | Zhangzhou | 24.45 | 117.7 | S | 4 | 12 | 1.250 | 1.147 | 0.000 | 0.068 | **1.000** |
| SG | Shaoguan | 24.65 | 113.53 | S | 7 | 20 | 1.583 | 1.259 | 0.004 | 0.177 | 0.975 |
| GL | Guilin | 25.13 | 110.66 | S | 3 | 9 | 1.500 | 1.365 | 0.028 | 0.221 | 0.800 |
| CH | Weining | 26.65 | 106.13 | S | 5 | 14 | 1.250 | 1.071 | 0.000 | 0.051 | **1.000** |
| KM | Kunming | 24.9 | 102.68 | S | 2 | 5 | 1.583 | 1.350 | 0.000 | 0.237 | **1.000** |
| EH | Dali | 25.8 | 100.2 | L | 12 | 25 | 1.833 | 1.525 | 0.003 | 0.288 | 0.988 |
| Overall | | | | | 132 | 357 | 5.000 | 1.965 | 0.013 | 0.439 | 0.976 |

Note:
The population size was labeled by S (Small, <500) or L (Large, >1,000); $N_a$, mean number of alleles; $A_e$, number of effective alleles; $H_o$, observed heterozygosity; $H_e$, expected heterozygosity; FI, fixation index. The FI values equaled to 1 are shown in bold font.

clustered into Group II (South group) (Fig. 2). This pattern was also supported by the phylogenetic trees (Fig. 3; Fig. S3). We also demonstrated the genetic divergence pattern illustrated by STRUCTURE when $K = 3$, where intermediate populations between the north and south groups were further clustered, showing genetic subdivisions within genetic groups (Fig. 2). The PCoA generated a similar pattern of population genetic structure. The first two components of PCoA explained a total variation of 39.52% (Fig. 4). The first principal coordinate accounted for 31.73% of the total variation and allowed the discrimination of the north and south genetic groups. The second coordinate, which accounted for 7.79% of the total variation, showed no clear differentiation pattern. The Mantel test provided significant evidence for isolation by distance (IBD) in *Z. latifolia* ($R^2 = 0.14$, $p < 0.01$) (Fig. 5).
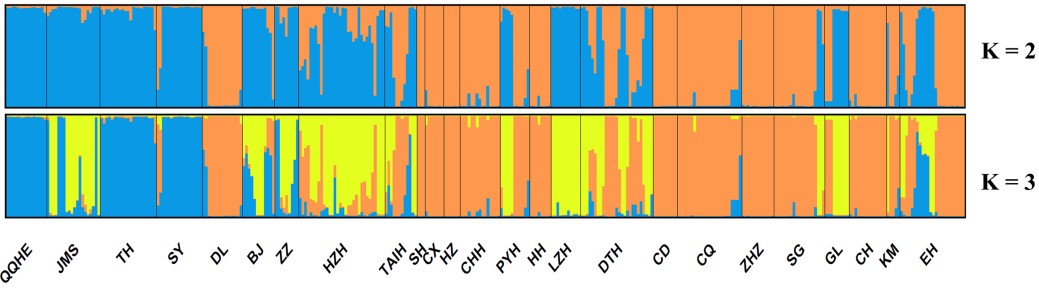

**Figure 2 Proportions of ancestry of 25 *Z. latifolia* populations based on *K* = 2 and *K* = 3 subdivisions.** Each bar represents the proportional membership assignments of an individual.

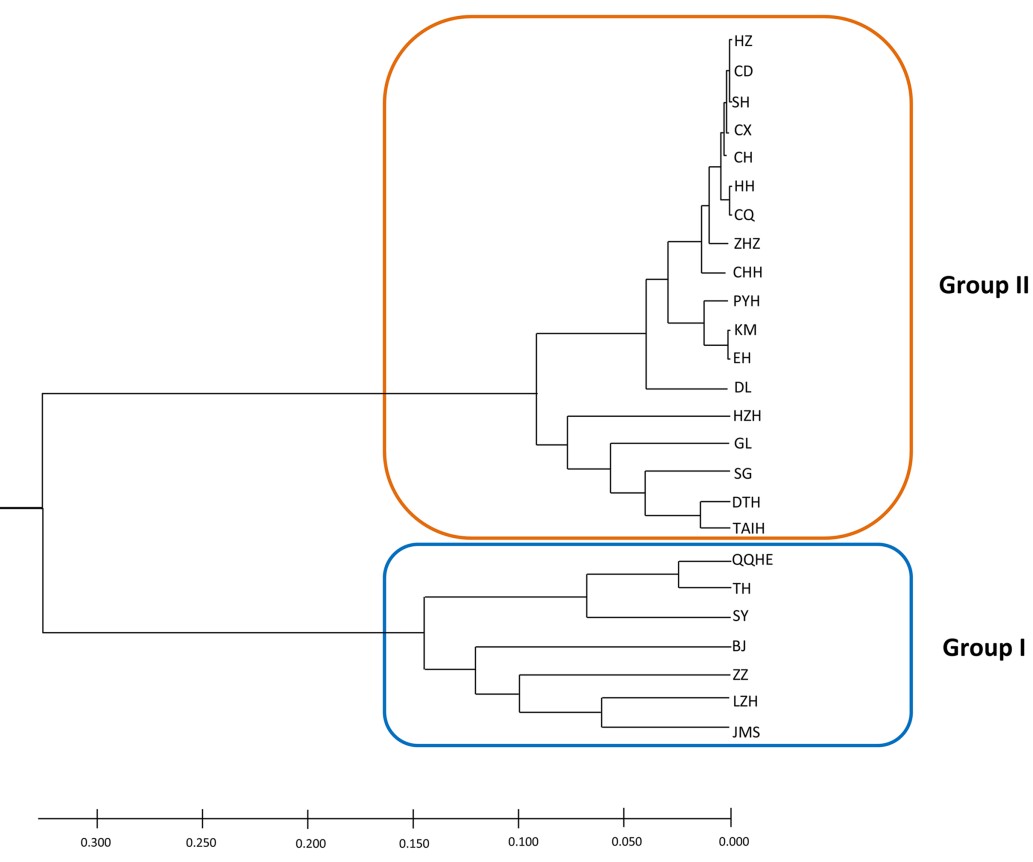

**Figure 3 The phylogeny tree (UPGMA) for the 25 wild *Z. latifolia* populations.** Two genetic groups (Group I and Group II) were illustrated.

## Development of a preliminary core collection

The CoreHunter3 software was utilized by setting the weights of MR to 0.5 each and applying sampling proportions of 10%, 15%, 20%, 25%, 30%, 40%, and 50%. This resulted in core germplasm samples of 36, 54, 71, 89, 107, 143, and 179, respectively. The sampling proportions in the SA method, which includes the SANA and SAGD methods, were kept consistent with those in CoreHunter3.

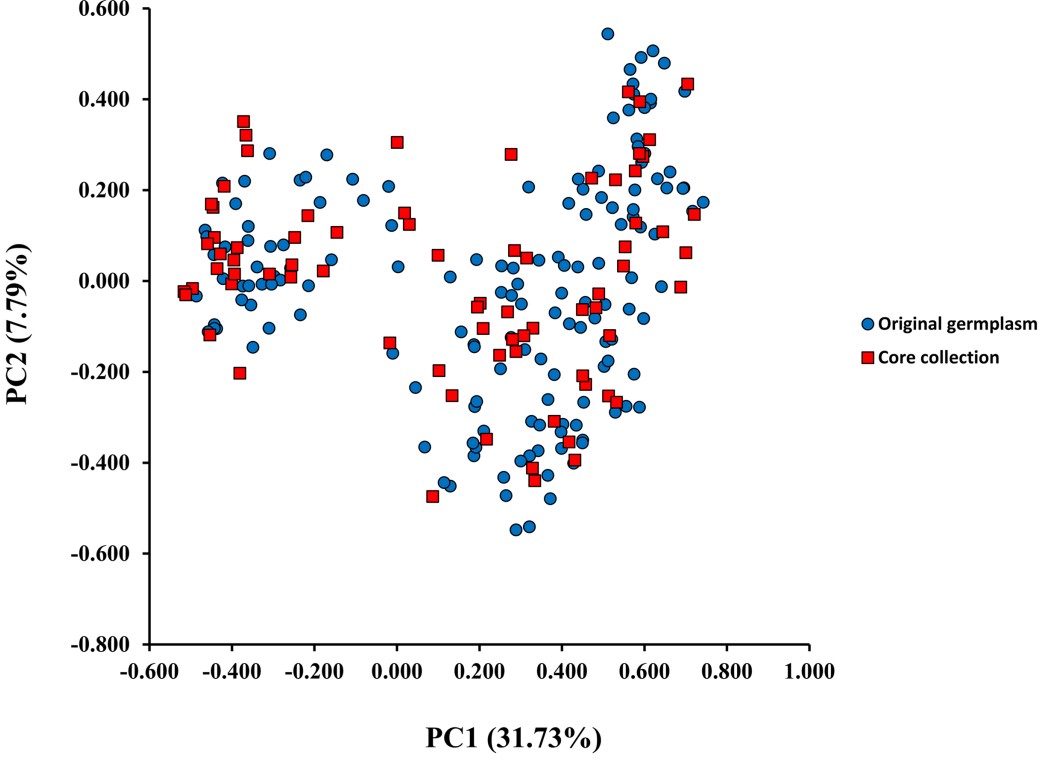

**Figure 4 The scatterplots of first two PC axis of principal-coordinate-analysis (PCoA) for original germplasm (blue circles) and core collection (red squares) of wild *Z. latifolia*.**

The diversity parameters for each sampled core collection are presented in Table 2. For SANA, the $H_e$ peaked at 0.444 when the core germplasm number reached 89 (25%), while the genetic diversity under SAGD increased linearly, peaking at 0.426 when the sample size was at its maximum. In contrast, the highest genetic diversity (0.582) was observed when the sampling proportion was set to 10% in CoreHunter3.

The core germplasms generated by different methods showed minimal overlap (Fig. 6). After removing redundant individuals, we obtained a merged core collection containing 92 accessions. Table 3 illustrates that there were no significant differences in genetic diversity parameters between the 92 core accessions and the original 357 accessions. Figure 3 also provides insight into the distribution of individuals as determined by principal coordinates analysis, showing that the 92 core accessions were evenly distributed across the entire set of 357 samples, indicating effective preservation of genetic diversity.

## DISCUSSION

### The distribution and habitat changes of *Z. latifolia* in China

According to the results of our nationwide survey of wild *Z. latifolia* populations during the 3-year period from 2019 to 2021, no wild *Z. latifolia* populations were found in 15 of the 40 survey areas that were pre-set based on the herbarium's historical distribution data. These areas are mainly located in northern and northwestern China. In contrast, wild

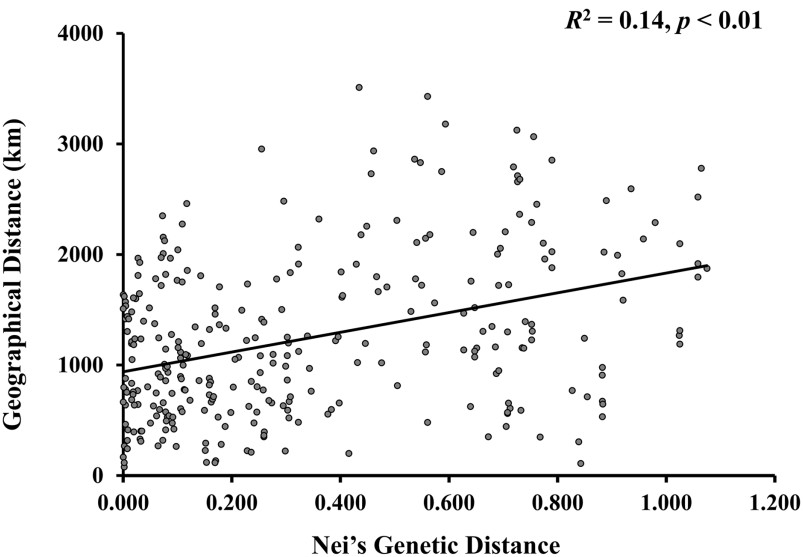

**Figure 5 The scatterplots between genetic distance and geographic distance for *Z. latifolia* populations.** Genetic distance is represented by pairwise Nei's distance among populations, which is correlated with geographic distance between populations. The regression line overlays the scatterplot (Mantel test, $R^2 = 0.14$, $p < 0.01$).

*Z. latifolia* populations were found in the remaining 25 survey areas. Currently, wild *Z. latifolia* is concentrated in the Yangtze River Basin, and some populations also exist in the Haihe River Basin and Northeast China. However, wild *Z. latifolia* populations in areas south of the Yangtze River Basin are difficult to find, which is consistent with the findings of *Zhao et al. (2018)*. Moreover, we found large habitat loss in less than 5 years when we revisited some populations, mainly due to the destruction of wetlands during urbanization. In fact, in recent decades, a large number of freshwater wetlands have been reclaimed and their areas drastically reduced due to factors such as population growth and a lack of understanding of the ecological functions of wetlands. As a result, suitable habitats for wild *Z. latifolia* have been lost. According to incomplete statistics, since the end of the 1940s, more than one-third of the lakes in the middle and lower reaches of the Yangtze River have been reclaimed, with a total area of more than 13,000 km², and over 1,000 lakes have disappeared due to reclamation. About two-thirds of the lakes in the middle reaches of the Yangtze River have vanished due to reclamation, with the area of Dongting Lake shrinking from 4,350 to 2,625 km², and the area of Poyang Lake decreasing from 5,200 to 2,933 km² (*Yang et al., 2010*). Additionally, the blockage of the water system by water conservancy projects, such as the construction of artificial gates and embankments, may further limit the natural expansion of wild *Z. latifolia* populations. For example, the middle and lower reaches of the Yangtze River, which harbor the largest and most concentrated wild populations of *Z. latifolia*, have only three lakes—Dongting Lake, Poyang Lake, and Shijiu Lake—that remain naturally connected to the river.

**Table 2 Genetic diversity parameters of wild *Z. latifolia* germplasms for different core subsets.**

| Method | Sampling proportion | Accession number | Population number | $A$ | $H_e$ |
|--------|--------|--------|--------|--------|--------|
| Overall | 100% | 357 | 25 | 60 | 0.439 |
| SANA | 10% | 36 | 17 | 42 | 0.422 |
| | 15% | 54 | 22 | 47 | 0.417 |
| | 20% | 71 | 22 | 48 | 0.412 |
| | 25% | 89 | 23 | 55 | 0.444 |
| | 30% | 107 | 24 | 49 | 0.405 |
| | 40% | 143 | 25 | 56 | 0.429 |
| | 50% | 179 | 24 | 56 | 0.437 |
| SAGD | 10% | 36 | 15 | 38 | 0.333 |
| | 15% | 54 | 18 | 44 | 0.382 |
| | 20% | 71 | 22 | 48 | 0.389 |
| | 25% | 89 | 21 | 50 | 0.402 |
| | 30% | 107 | 23 | 55 | 0.403 |
| | 40% | 143 | 23 | 55 | 0.415 |
| | 50% | 179 | 24 | 58 | 0.426 |
| MR | 10% | 36 | 11 | 54 | 0.582 |
| | 15% | 54 | 14 | 56 | 0.569 |
| | 20% | 71 | 15 | 59 | 0.546 |
| | 25% | 89 | 19 | 60 | 0.546 |
| | 30% | 107 | 20 | 60 | 0.534 |
| | 40% | 143 | 21 | 60 | 0.515 |
| | 50% | 179 | 25 | 60 | 0.497 |

**Note:**
$A$, allele number; $H_e$, expected heterozygosity.

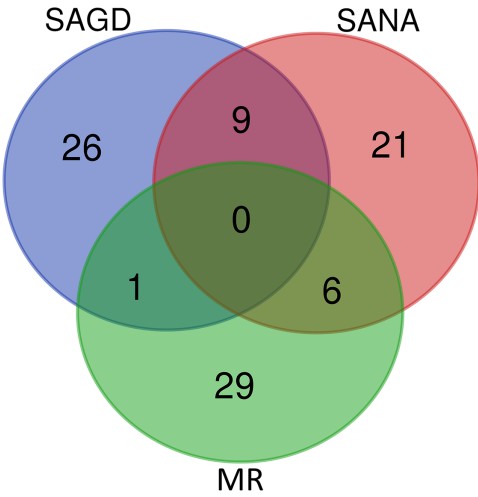

**Figure 6 Venn diagram of core germplasms identified by three different methods (SAGD, SANA and MR).**

**Table 3 Comparison of genetic parameters between core collection and original germplasm.**

| Parameters | Original germplasm | Core germplasm |
|---|---|---|
| $N$ | 357 | 92 |
| $A$ | 60 | 57 |
| $A_e$ | 1.965 | 2.177 |
| $I$ | 0.345 | 0.917 |
| $H_o$ | 0.013 | 0.016 |
| $H_e$ | 0.439 | 0.490 |
| $FI$ | 0.976 | 0.971 |

Note:
$N$, sample size; $A$, allele number; $A_e$, number of effective alleles; $I$, Shannon Index; $H_o$, observed heterozygosity; $H_e$, expected heterozygosity; $FI$, fixation index.

## Genetic diversity of wild *Z. latifolia* populations

Genetic diversity is an important component of biological diversity, as richer genetic diversity within a species enhances its potential ability to adapt to environmental changes (*Hughes et al., 2008*). As a perennial aquatic plant, *Z. latifolia* exhibits strong clonal growth (*Xu et al., 2008*; *Chen et al., 2012*). We observed this characteristic during the sampling process and took necessary measures to avoid collecting too many clones. The phylogeny of the genus *Zizania* suggests that the ancestors of *Z. latifolia* originated in the Americas and migrated to the Far East and East Asia through the Bering Strait during the Tertiary period (*Xu et al., 2010*). *Xu et al. (2008)* investigated the phylogeny and genetic structure of *Z. latifolia* in China based on the nuclear gene sequence *Adh1a*, finding that populations at high latitudes (Northeast China) possessed a richer range of haplotypes. *Xu et al. (2015)* also used three pairs of universal microsatellite markers to compare the genetic diversity of four species within the genus *Zizania*. They found that the genetic diversity of *Z. latifolia* was relatively low ($H_e = 0.374$) compared to *Z. palustris* ($H_e = 0.630$). However, this work may have underestimated genetic diversity due to the use of too few molecular markers. Subsequently, *Zhao et al. (2019)* and *Wagutu et al. (2022)* examined the genetic diversity and genetic structure of wild populations of *Z. latifolia* in China using SSR molecular markers. Both studies found high genetic diversity (mean diversity: $H_e = 0.426$ in *Zhao et al., 2019*, $H_e = 0.173$ in *Wagutu et al., 2022*; overall diversity: $H_e = 0.432$ in *Zhao et al., 2019*, $H_e = 0.241$ in *Wagutu et al., 2022*). In comparison, the present study found a similar level of overall genetic diversity ($H_e = 0.439$) as *Zhao et al. (2019)*, but higher than *Wagutu et al. (2022)*; and the mean diversity ($H_e = 0.207$) was at the same level as Wagutu. These findings may be due to sampling differences between studies. Although *Wagutu et al. (2022)* surveyed more populations, their sampling was limited to watersheds on five latitudinal gradients. Populations within the watersheds may trend to be genetically identical due to the presence of gene flow, thus making relatively low overall genetic diversity. In contrast, both *Zhao et al. (2019)* and our work sampled populations extensively across the country. Unfortunately, previous studies (*Xu et al., 2015*; *Zhao et al., 2019*; *Wagutu et al., 2022*) did not calculate the genetic parameters of $I$ and $PIC$, preventing us from directly assessing differences in genotypic diversity across studies. However, the very low observed

heterozygosity ($H_o$) coupled with an unusually high fixation index ($FI$) suggests a heterozygote deficiency in most populations. To collect more representative genotypes for constructing a germplasm repository, we sampled an area (population) with multiple sample sites spread far apart to cover as much of the area as possible, spacing samples approximately 1,000 m apart within each sample point. This sampling strategy may have two potential consequences: First, multiple populations might have been mistakenly sampled as one, leading to genetic structure subdivision and high overall genetic diversity (Wahlund effect). Second, if some wild populations of *Z. latifolia* are already experiencing heterozygote deficiency (*Zhao et al., 2019*), the sign of heterozygote deficiency could be further amplified because of the higher likelihood of collecting homozygous individuals. The increased probability of biparental inbreeding due to extensive clonal growth in *Z. latifolia* may also raise the proportion of homozygous genotypes in the population. Additionally, heterozygote deficiency may result from the insufficient polymorphism and number of molecular markers used in this study. For example, *Wagutu et al. (2022)* used 46 pairs of SSR markers and found heterozygote excess ($FI = -0.245$), whereas only 12 pairs of SSR markers were used in our study. Although conventional gel electrophoresis may underestimate the actual number of alleles, the relatively low PIC (mean PIC = 0.383) also suggests that this earlier batch of developed SSR markers used in our study may have insufficient polymorphism and that new high-polymorphic SSR markers need to be developed using the genome or transcriptome data (*Quan et al., 2009*; *Wagutu et al., 2022*; *Wang et al., 2023*). Now that a high-quality chromosome-level genome of *Z. latifolia* has been released (*Yan et al., 2022*), subsequent studies can employ whole-genome resequencing to more accurately evaluate the genetic diversity of *Z. latifolia* (*Zou et al., 2024*).

## Genetic population structure of wild *Z. latifolia*

The results of this study also showed high genetic differentiation ($F_{st} = 0.452$) among *Z. latifolia* populations in China, suggesting restricted gene flow among these populations. Specifically, we found significant isolation by distance (IBD) through the Mantel test, and STRUCTURE, PCoA, and UPGMA clustering analyses indicated that the wild *Z. latifolia* populations can be divided into north and south genetic groups. The natural habitat of *Z. latifolia* is generally fragmented, forming isolated habitats that limit gene exchange between populations. This aligns with the observation that aquatic plants typically exhibit high genetic differentiation (*Barrett, Eckert & Husband, 1993*). Even though these habitats may sometimes achieve intermittent connectivity through water regimes, this connectivity does not necessarily coincide with the seed maturity period required for effective gene flow *via* waterborne seed dispersal. Additionally, whether birds are significant mediators of seed dispersal in *Z. latifolia* remains unclear, and pollen flow is only effective in localized areas. Overall, *Z. latifolia* lacks the ability to disperse gene flow over long distances, resulting in isolation by distance on a large scale (*Willson, 1983*). Furthermore, the genetic structure of aquatic plants is influenced not only by geographic isolation but also by the connectivity of water systems (*Wang et al., 2008*; *Honnay et al., 2010*; *Davis et al., 2018*). Both *Zhao et al. (2019)* and *Wagutu et al. (2022)* have previously reported a north-south genetic differentiation in wild populations of *Z. latifolia*, and our results also support this

conclusion. However, *Wagutu et al. (2022)* concluded that populations across different latitudinal gradients were more genetically similar within watersheds, whereas genetic subdivision occurred between watersheds in a latitudinal direction. In contrast, *Zhao et al. (2019)* did not observe latitudinal genetic subdivision, instead pointing out genetic subdivision in an east-west direction within the Yangtze River Basin. To further explore these patterns, we analyzed the genetic structure of *Z. latifolia* populations using STRUCTURE software by applying $K = 3$ (Fig. 2). However, we did not find any clear genetic subdivisions associated with the geographic distribution of the populations. Interestingly, in both STRUCTURE and UPGMA analyses, the DL populations located in high northern latitudes were clustered with populations in the south, raising questions about the origin of this population that should be addressed in subsequent work. *Zhao et al. (2019)* has reported the existence of *Z. latifolia* populations formed by the escape of cultivars into natural habitats. If feral *Z. latifolia* samples were incorrectly collected, this could lead to misclassification of the genetic structure of the populations.

## The designing of a preliminary core collection for wild *Z. latifolia*

Core germplasm is a key component in plant breeding that can increase the effectiveness of utilizing plant genetic resources for genetic improvement and new variety development. The purpose of core germplasm is to maximize the representation of the genetic diversity of the entire species with a minimum number of genetic resources. To accomplishing this goal, the sampling strategy is important, and it determined the results of core germplasm construction. A suitable sampling plan can optimize the preservation of genetic diversity while minimizing genetic redundancy. Of these, stratified sampling is often considered superior to random sampling in many aspects, as demonstrated by numerous studies (*Brown, 1989*; *Casler, 1995*; *Escribano, Viruel & Hormaza, 2008*; *Kumar et al., 2020*). Stratified sampling can be implemented using various methods, including proportional, logit, square root, and genetic diversity-based sampling, each with its own advantages and disadvantages. When selecting a sampling strategy, it is important to consider factors such as the amount of germplasm material, grouping, and genetic diversity.

Core germplasm construction involves extracting a portion of the total samples from existing germplasm resources that meet specific criteria. Achieving this goal hinges on selecting appropriate sampling methods and determining suitable sampling ratios. Currently, the commonly used software for constructing core germplasm includes PowerMarker, PowerCore, and CoreHunter. The proportion of core germplasm relative to the original materials in various plants, both domestically and internationally, ranges from 5% to 40% (*Wang et al., 2007*; *Escribano, Viruel & Hormaza, 2008*; *Liang et al., 2015*; *Dervishi et al., 2021*). However, none of these studies provided a uniform sampling rate, primarily due to differences in sampling completeness, levels of genetic diversity, and the biological characteristics specific to each plant. In our research, we used PowerMarker v3.25 (SANA and SAGD) and CoreHunter3 (MR) to construct a preliminary core collection of wild *Z. latifolia*. Our results revealed very limited intersection between the core germplasm constructed by different methods using a 10% sampling ratio (Fig. 6), likely due to differences in the algorithms underlying each method. As a result, we

combined the core germplasm constructed by each of the three methods. Evaluating the representativeness of core samples in relation to the diversity of the entire germplasm resource is also a key focus in core germplasm studies. Referring to previous studies, we first created a principal component map based on the final core germplasm and compared it with the principal component distribution maps of all materials, finding a high degree of agreement (Fig. 4). Additionally, no significant differences were observed between the genetic diversity indices of the final core germplasm and the original germplasm materials (Table 3). The results demonstrated that the final core germplasm was well-matched with the original germplasm, indicating the rationality of the final core germplasm screening results. This core collection will facilitate the utilization of *Z. latifolia* in developing new crop varieties with enhanced yield and resistance traits.

## CONCLUSIONS

The present study addresses the importance of genetic diversity assessment and core collection development for the conservation of *Z. latifolia*, a significant genetic resource for rice improvement. Despite the limited number of molecular markers and relatively insufficient sample size, our findings highlight the impact of habitat loss and fragmentation on the genetic structure of wild populations, with a clear north-south genetic differentiation pattern. The successful design of a preliminary core collection, using various sampling strategies, ensures the efficient representation of genetic diversity and provides a foundation for future breeding and genetic research. Moreover, the insights gained from this study call for immediate conservation actions to protect the remaining wild populations of *Z. latifolia* from further decline due to anthropogenic pressures. Future research should leverage whole-genome resequencing to refine the assessment of genetic diversity, considering the recent availability of a high-quality chromosome-level genome of *Z. latifolia*.

## ACKNOWLEDGEMENTS

We express thanks to Jianli Wu and Huimei Wang providing for critical reading and helpful suggestions on improving our manuscript.

### Funding

This work was supported by the National Natural Science Foundation of China (Grant Nos. 31460378, 31600293, 32260091), Natural Science Foundation of Jiangxi Province (Grant No. 20212BAB205029) and China Agricultural Research System (No. CARS-01-60). The funders had no role in study design, data collection and analysis, decision to publish, or preparation of the manuscript.

### Grant Disclosures

The following grant information was disclosed by the authors:
National Natural Science Foundation of China: 31460378, 31600293, 32260091.

Natural Science Foundation of Jiangxi Province: 20212BAB205029.
China Agricultural Research System: CARS-01-60.

## Competing Interests

The authors declare that they have no competing interests.

## Author Contributions

- Xiangliang Lei performed the experiments, analyzed the data, prepared figures and/or tables, and approved the final draft.
- Xiaona Su performed the experiments, analyzed the data, prepared figures and/or tables, and approved the final draft.
- Chengchuan Zhou performed the experiments, prepared figures and/or tables, and approved the final draft.
- Shaolin Jiang conceived and designed the experiments, authored or reviewed drafts of the article, and approved the final draft.
- Xiaoquan Yuan performed the experiments, authored or reviewed drafts of the article, and approved the final draft.
- Yao Zhao conceived and designed the experiments, analyzed the data, prepared figures and/or tables, and approved the final draft.
- Shaomei Jiang conceived and designed the experiments, authored or reviewed drafts of the article, and approved the final draft.

## Data Availability

The raw data are available in the Supplemental Files and at Figshare: Zhao, Yao; Lei, Xiangliang (2024). Lei et al. 2024 SSR raw data of *Zizania latifolia*. figshare. Dataset. https://doi.org/10.6084/m9.figshare.26384773.v1.

## Supplemental Information

Supplemental information for this article can be found online at http://dx.doi.org/10.7717/peerj.18909#supplemental-information.

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
