# Peer review of "Genetic structure and designing a preliminary core collection of Zizania latifolia in China based on 12 microsatellites markers"

_PeerJ, doi:10.7717/peerj.18909_

## Round 0.1 · original submission · Major Revisions

Dear colleagues, your manuscript has been evaluated by expert reviewers and it is my opinion that has merit for publication after revision.

Please address these comments in the revised version.

Best regards,
Dr Nikolaos Nikoloudakis

Reviewer 1 ·

Basic reporting

This manuscript assessed the genetic diversity of a large Zizania latifolia collection to identify a core collection of germplasm for future research and breeding efforts. Overall, the manuscript is well written and descriptive. Below are a few minor comments.

Introduction – The background knowledge provided was explained well but it is quite long to read. I would suggest the authors edit this section and write more succinctly.

M&M
- Lines 136-139. How many individual plants – 357? What are sampling points?

Experimental design

very well done

Validity of the findings

very well done

·

Basic reporting

1. Title: The species name Zizania latifolia should be in italics.
2. Lines 44-45: The reference to the FAO should be accompanied by the year of the data.
2. Lines 44-45: The reference to the FAO should be accompanied by the year of the data.
3. Literature Survey: The manuscript lacks discussion on previous studies of Z. latifolia genetic diversity using molecular markers such as SNPs. Please incorporate this for a more comprehensive background.

Experimental design

4. Sampling Period and Environmental Factors:
The sampling was conducted between 2019 and 2021. Were there significant environmental changes, such as extreme weather events or habitat alterations, that could have influenced the genetic diversity and population structure during this period? Please elaborate on how these factors were considered in your analysis.

Validity of the findings

5. Consider adding a figure (either in the manuscript or as supplementary material) showing a polyacrylamide gel image that depicts the amplification profile of some SSR markers.
6. For the AMOVA test results, it would be helpful to include a figure or table to visually support the discussion of the findings.
7. In Figure 5 (Mantel test between genetic and geographic distances), the R² value (0.14) and p-value (p < 0.01) should be clearly presented in the figure for completeness.
8. UPGMA Dendrogram: I recommend including a UPGMA dendrogram of the 357 accessions based on the 12 SSR markers for more detailed resolution and clarity.
9. Genetic Diversity: The genetic diversity (He = 0.439) reported in the study aligns with previous findings but shows variations compared to Wagutu et al. (2022). Could you provide an explanation for these differences, especially in relation to the varying number of SSR markers used? Additionally, why were only 12 SSR markers chosen for the analysis, and would including more markers have strengthened the genetic diversity estimates?
10. Core Collection Validation: The manuscript mentions that the core collection is representative of the original germplasm. Could you clarify the specific metrics or validation techniques used to confirm this? Was any independent verification or comparison made with other core collection methods from similar studies?

·

Basic reporting

The article is generally well-written, with no significant comments regarding the language or the structure. However, there are important issues concerning the figures and references that need to be addressed. Specifically, the label for the inferred K values of Figure 2, should be indicated on the figure to enhance understanding of the results. Additionally, the reference used for the marker Zt23 needs to be corrected; it is currently cited that it is from Richards et al. 2004, but it should refer to the 2007 paper from the same author.

Experimental design

The authors present a well-defined research question that addresses a relevant knowledge gap. However, several aspects could be improved in the article. Firstly, the criteria for selecting the molecular markers are not adequately detailed. It would strengthen the manuscript if the authors clarified the selection process, specifying why they chose these particular SSRs over others. For instance, the authors referenced the work of Wagutu et al. (2022), which used 46 SSRs. While some of these SSRs overlap with those used in this study by Quan et al. (2009) and Richards et al. (2007), Wagutu et al. also introduced new markers that could have provided additional insights if included here.
Furthermore, the manuscript would benefit from including calculations of the polymorphism information content (PIC) for each primer set to more rigorously assess marker informativeness and variability. This additional information would enhance replicability and provide a more comprehensive evaluation of the marker utility.

Validity of the findings

The manuscript contributes valuable insights into the study of Z. latifolia diversity. However, several points should be reviewed. In the Results section (line 210), there is an error in the reference to Figure 2, which is not a phylogenetic tree but instead displays the genetic group structure based on K values; this figure should be removed from the brackets to avoid confusion. Additionally, in the Discussion section (lines 351 and 357), the software name "PowerMarker" is misspelled and should be corrected for clarity.

·

Basic reporting

Lei et al. used 12 SSR to determine the genetic diversity and population structure of Zizania latifolia. They did a very good job. Analysis are correct and the paper is well-written. However, I consider minore correction is needed. Please see below.

• Abstract
 Indicate number of populations evaluated

• Material and Methods
 Each survey area is a being considered a population?
 Was DNA quality and quantity assessed? Please include details
 How did the authors select the 24 SSR? Please include details
 Indicate more details about the final dataset. For exampe, was it a presence/absence data set? What were the dimensions of the data set?
 How many Ks were considered for the STRUCTURE analysis?

• Results
 Shannon’s Index is not mentioned in the “Genetic diversity” subsection.
 Raw data is not cited in the manuscript
 In the abstract the authors mention “moderate genetic diversity (HE = 0.439), but in the result section they indicate genetic diversity is high. Please clearly indicate HE is high.
 Figures S1 indicates K from 1 to 19 were considered. However, the authors collected samples from 25 populations. So, at least K should have been considered from K =1 to K = 25.
 Please indicate where this germplasm is maintained. Germplasm bank?
 Fig. 1. Indicate latitude, longitude
 Fig. 3. Please indicate what the acronym mean
• Discussion
 Briefly indicate future work with this germplasm
 Indicate limitations of the study. Perhaps number of SSR?

Experimental design

It is OK

Validity of the findings

Important

---

## Round 0.2 · Minor Revisions

Dear colleague. The reviewers have now completed their positive assessment of your manuscript and I also believe that it can be accepted after the minor revision asked by reviewer 3

Reviewer 1 ·

Basic reporting

Changes are sufficient for publication

Experimental design

Meets standards

Validity of the findings

Changes are sufficient for publication

·

Basic reporting

Nice work of the revision

No comment

Experimental design

No comment

Validity of the findings

No comment

·

Basic reporting

No comment

Experimental design

Lei et al. did a perfect job incorporating the reviewers' changes to clarify some doubts in the paper. I appreciate that they explained the markers' selection and the calculation of PIC values. However, in this latter step, they only added the values without discussing the results. They should compare these results with other studies, as some SSR markers should likely be excluded from the analysis due to their low PIC values since it provides an estimate of the discriminatory power of the locus. The same applies to Shannon’s Index. These results should be compared with other studies, as this index also estimates genetic diversity.

Validity of the findings

No comment

---

## Round 0.3 · accepted · Accept

Dear colleagues, following your revision, I recommend acceptance of the manuscript as it stands